

# A top-down evaluation of bottom-up estimates to reduce uncertainty in methane emissions from Arctic wetlands

Luana S. Basso[1], Goran Georgievski[2], Victor Brovkin[2], Christian Beer[3], Christian Rödenbeck[1], Mathias Göckede[1]

[1]Department of Biogeochemical Signals, Max Planck Institute for Biogeochemistry, Jena, 07745, Germany

[2]Department of Climate Dynamics, Max Planck Institute for Meteorology, Hamburg, 20146, Germany

[3]Department of Earth System Sciences, University of Hamburg, Hamburg, 20146, Germany

*Correspondence to*: Luana S. Basso (lbasso@bgc-jena.mpg.de)

## Abstract.

Wetlands are a major natural source of atmospheric $CH_4$, however, accurately estimating their emissions is difficult due to the complex biogeochemical interactions and spatial heterogeneity of wetland environments. This study explores how a combination of atmospheric inverse and process-based modelling can reduce the discrepancy in Arctic wetland estimates between bottom-up and top-down approaches. We employed the Jena CarboScope global inversion system, incorporating prior wetland fluxes simulated by the JSBACH land surface model, which is part of the Max Planck Institute Earth System Model (MPI-ESM). We conducted a series of inversion experiments, each incorporating JSBACH-generated $CH_4$ fluxes based on different $CH_4$ production $Q_{10}$ values to test the temperature sensitivity of emissions. Additionally, we examined the impact of changing the baseline $f_{CH_4}$ fraction value, which defines the fraction of anaerobically mineralized carbon converted to $CH_4$, while keeping all other JSBACH and inversion settings constant. Our findings show that, at a pan-Arctic scale, using a $CH_4$ $Q_{10}$ value of 1.8 produces the best agreement between the two approaches. However, no single $Q_{10}$ value yielded optimal agreement between the simulated fluxes and the fluxes inferred from atmospheric observations across all subregions. Instead, the best performance varied spatially, with different $CH_4$ production $Q_{10}$ values and baseline $f_{CH_4}$ fraction leading to a better flux agreement in specific areas. These results highlight the importance of using regionally specific parameters to more accurately estimate wetland $CH_4$ emissions, and the potential of employing atmospheric inversions to guide bottom-up process models towards regionally representative parameter settings.

## 1. Introduction

Methane ($CH_4$) is the second most important anthropogenic greenhouse gas and it is emitted from both natural and anthropogenic sources. Combined wetlands and inland freshwaters are the largest natural source of $CH_4$ to the atmosphere, accounting for about 28-37% (by bottom-up and top-down estimates, respectively) of the global total $CH_4$ emissions (Saunois et al., 2025). However, quantifying these emissions remains challenging due to the complexity of biogeochemical processes and the spatial variability of these ecosystems. Process-model ensemble estimates indicate that, between 2010 and 2020,





wetlands emitted approximately $158 \pm 24$ TgCH$_4$ y$^{-1}$. This represents an increase of ~5 TgCH$_4$ y$^{-1}$ compared to the 2000-2009
average, with the most substantial increases observed in tropical regions, followed by mid- and high-latitude areas (Zhang et
al., 2025).
Global and regional CH$_4$ emissions are estimated using both bottom-up or top-down approaches. Bottom-up methods,
including data-driven ecosystem flux upscaling and process-based models, provide detailed information with fine-scale
resolution for both, processes and spatial heterogeneity. Process-based models simulate CH$_4$ emissions by mathematically
representing ecosystem dynamics, biogeochemical cycles, and physical processes. Nevertheless, it is challenging to extrapolate
these estimates to regional or global scales because wetland characteristics (e.g., extent, hydrology and vegetation) vary
substantially across space, and simulated CH$_4$ fluxes are highly sensitive to the choice of model parameterizations. Mechanistic
modeling of net surface CH$_4$ emissions requires capturing a range of complex, interacting processes (Conrad, 1999; Moser et
al., 2025; Riley et al., 2011).
Anaerobic CH$_4$ production is the result of a number of biogeochemical processes that take place in a chain or in parallel
(Conrad, 2020; Moser et al., 2025; Song et al., 2020). After an enzymatic breakdown of macromolecules, fermentation of the
resulting dissolved organic matter (DOC) leads to acetate, hydrogen and CO$_2$. In either acetoclastic or hydrogenotrophic
methanogenesis, these byproducts are immediately further used to finally produce CH$_4$ and CO$_2$ (Conrad, 2020). In addition,
alternative electron acceptors, such as Fe-III can be utilized by microbes to produce CO$_2$ from acetate (Sulman et al., 2022;
Zheng et al., 2019). The net CH$_4$:CO$_2$ production ratio is therefore determined by the relative importance of these underlying
processes, which in turn are dependent on environmental conditions. That is why in laboratory incubation experiments, a large
range of this production ratio has been observed (Knoblauch et al., 2018). After production, CH$_4$ may be consumed by
methanotrophic bacteria (Knoblauch et al., 2015; Riley et al., 2011) or transported to the atmosphere via plant aerenchyma,
ebullition, or diffusion through soil or water (Kaiser et al., 2017; Walter and Heimann, 2000; Wania et al., 2010). That leads
to a CH$_4$:CO$_2$ emission ratio at the surface which is different from the CH$_4$:CO$_2$ production ratio. Since underlying
biogeochemical processes are very complex and dependent on detailed environmental conditions, global-scale land surface
models usually represent anaerobic CH$_4$ production as a first-order decay of soil organic matter with adjusted rate constants.
And then, a fixed ratio of CH$_4$ versus CO$_2$ production out of that decomposition is applied (Guimberteau et al., 2018; Kleinen
et al., 2020; Moser et al., 2025; Ricciuto et al., 2021; Sellar et al., 2019). Here, the models can differ in whether the ratio
applies to the CH$_4$ production or emission. The JSBACH v3.2 (Reick et al., 2021) that we apply in this study is taking the first
approach and mechanistically distinguish between methanogenesis and methanotrophy.
Developing these models requires balancing the inclusion of key mechanisms with limitations such as structural and
parameter uncertainty, spatial heterogeneity, sparse observational data, uncertain initial and boundary conditions, and
computational constraints (Riley et al., 2011). Previous studies have shown that CH$_4$ emissions are highly sensitive to
parameters regulating microbial production and oxidation processes (Chinta et al., 2024; Riley et al., 2011; Song et al., 2020).
A higher CH$_4$:CO$_2$ ratio indicates a greater dominance of CH$_4$ in production and emission relative to CO$_2$ (Chinta et al., 2024).
Based on anaerobic incubations of thermokarst lake sediments, Gonzalez Moguel et al. (2025) observed that the $\Delta^{14}$C values





of both $CH_4$ and $CO_2$ showed strong positive correlations with net $CH_4$ production rates and $CH_4$:$CO_2$ ratios. This indicates
that $CH_4$ production occurs faster and at a higher rate when younger organic matter decomposes. These patterns suggest that
the presence of younger carbon substrates increases methanogenesis compared to overall fermentation and anaerobic
respiration (Gonzalez Moguel et al., 2025). A higher $CH_4$ production $Q_{10}$ indicates that $CH_4$ production increases more rapidly
with rising temperatures. This can indirectly enhance diffusive fluxes by creating larger concentration gradients between the
soil and the atmosphere (Chinta et al., 2024). However, as regional model sensitivity varies and site-specific measurements
may not be representative across broader areas, $CH_4$ production $Q_{10}$ are uncertain at large spatial scales. For example,
increasing $CH_4$ production $Q_{10}$ in high-latitude regions can reduce simulated $CH_4$ emissions by more than half, because the
temperature-dependent component, scaled relative to a reference temperature of 295 K, leads to a decline in $CH_4$ production
rate at the lower temperatures typical of these regions (Riley et al., 2011).  In contrast, the opposite pattern is observed in
tropical regions (Riley et al., 2011). Many large-scale land surface models still rely on simplified, fixed $CH_4$ production
fractions, which limits their ability to accurately represent observed spatiotemporal variability in $CH_4$:$CO_2$ production ratios
across Arctic landscapes (Moser et al., 2025). These differences in model structure, parameterization and initialization
contribute strongly to relative high uncertainties in wetland estimates (Poulter et al., 2017).
In JSBACH v3.2, anaerobic decomposition and $CH_4$ oxidation are temperature dependent. However, in addition to that,
the $CH_4$:$CO_2$ production ratio is also assumed to follow a $Q_{10}$ temperature sensitivity (Kleinen et al., 2020). That means that
we assume that the relative importance of the above-mentioned underlying biogeochemical processes changes in space and
time depending on the soil temperature. In addition, making the $CH_4$:$CO_2$ production ratio temperature dependent allows us to
additionally tune $CH_4$ versus $CO_2$ production across bioclimatic zones. One big research question now is, how high should be
the $Q_{10}$ value for this temperature dependency of the $CH_4$:$CO_2$ production ratio? In order to answer such question, we employ
a novel integration of bottom-up and top-down approaches.
Top-down approaches estimate net surface-atmosphere $CH_4$ fluxes using atmospheric observations (in situ, flask and/or
satellite measurements) in combination with prior flux information (from process-based models and/or inventories), and
atmospheric transport and chemistry models to link surface sources with atmospheric observations. Their ability to provide
accurate estimates of net surface-atmosphere fluxes is limited by sparse observational coverage, particularly in remote regions,
as well as by uncertainties in atmospheric transport, prior flux estimates, and atmospheric $CH_4$ sink processes  (Houweling et
al., 2017). These limitations can lead to significant uncertainties in the magnitude and spatial distribution of inferred emissions,
which makes attributing fluxes to specific sources or processes challenging. Still, despite these limitations, the inverse
modeling approach allowed us to derive important constraints on the global sources and sinks of $CH_4$ (Houweling et al., 2017).
Substantial discrepancies exist between bottom-up and top-down estimates of $CH_4$ emissions. From 2010 to 2019, top-
down approaches estimated global $CH_4$ emissions at 575 $TgCH_4$ $y^{-1}$ (553-586 $TgCH_4$ $y^{-1}$), whereas bottom-up estimates were
approximately 15% higher, at 669 $TgCH_4$ $y^{-1}$ (512-849 $TgCH_4$ $y^{-1}$) (Saunois et al., 2025). These differences, despite the fact
that bottom-up results are used as prior in top-down approaches, point to additional constraints of bottom-up $CH_4$ flux estimates
by atmospheric observations. For example, important large-scale $CH_4$ uptake by upland soils (Juncher Jørgensen et al., 2024;



Voigt et al., 2023) is usually underrepresented in land surface models (D'Imperio et al., 2023; Song et al., 2024). More
generally, we assume that bottom-up approaches are still very limited in their ability to upscale the complex and spatially
varying processes underlying $CH_4$ emissions. In boreal regions, inland freshwater sources dominate $CH_4$ emissions, accounting
for 41% and 54% in top-down and bottom-up budgets, respectively (Saunois et al., 2025). Similarly, Hugelius et al. (2024)
reported substantial discrepancies between bottom-up and top-down $CH_4$ emission estimates for the Arctic–boreal region, with
50 $TgCH_4$ $y^{-1}$ (29-71 $TgCH_4$ $y^{-1}$) for bottom-up and 20 $TgCH_4$ $y^{-1}$ (15-24 $TgCH_4$ $y^{-1}$) for top-down. Despite recent efforts to
improve monitoring networks and modeling frameworks, significant discrepancies remain between these approaches. Still,
top-down approaches can be used to assess the representativeness of bottom-up fluxes and their underlying parameterizations
on a large scale. Combining information from both methods can therefore help to reconcile discrepancies and improve the
consistency of $CH_4$ emission estimates at different spatial scales.
This study explores the use of atmospheric inverse modeling to constrain bottom-up estimates of wetland $CH_4$ emissions
in the Arctic-Boreal region. Using the Jena CarboScope global inversion system, we employed prior fluxes from the JSBACH
land surface model (a component of the MPI Earth System Model) and systematically varied key parameters that govern $CH_4$
production. Specifically, we tested a range of $Q_{10}$ values, which define the temperature sensitivity of $CH_4$ production, and
different $f_{CH_4}$ baseline values, which determine the proportion of anaerobically mineralized carbon converted to $CH_4$. We kept
other model settings constant throughout these tests. Integrating these parameter sensitivity experiments into the inversion
framework allowed us to assess which parameterizations yield the most consistent fluxes with atmospheric observations. This
approach enables us to identify regionally representative parameter settings and guide parameterizations that could improve
the consistency between bottom-up process models and top-down constraints on Arctic-Boreal wetland $CH_4$ emissions.
**2.  Methods**
**2.1.  Region and time period of interest**
Our Arctic-Boreal domain was defined based on The Boreal-Arctic Wetland and Lake Dataset – BAWLD (Olefeldt et al.,
2021), and we divided this region into 6 sub-regions for more detailed spatial analyses (Alaska, western Canada, eastern
Canada, Europe, western Russia, eastern Russia, Fig. 1). In recent decades, the atmospheric observation network suitable for
inverse modeling has expanded across the Arctic, with a considerable increase in available sites after 2010 (Vogt et al., 2025).
However, due to data-sharing disruptions associated with the ongoing conflict involving Russia and Ukraine, observational
data from Russian stations has been limited since 2022. Consequently, this study focuses on the period from 2010 to 2021,
when data coverage was more consistent across the full domain.





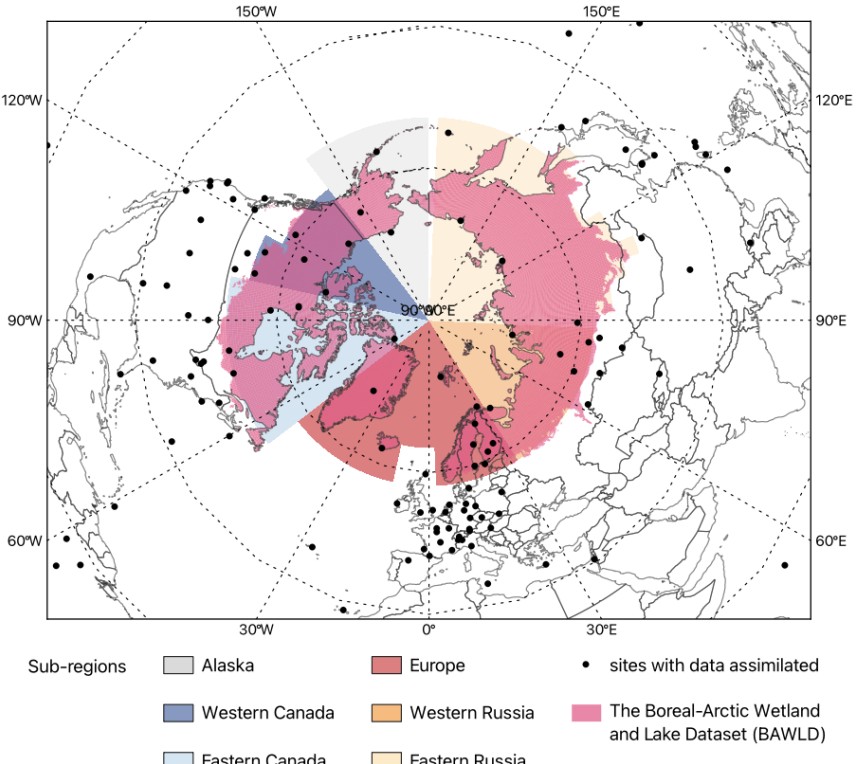

Figure 1: Geographic distribution of surface sites operated by different network providers where flask-based and/or continuous in-situ $CH_4$ measurements are available for assimilation into the inverse model (black dots). The colored boxes delineate the Arctic-Boreal regions (Alaska, western Canada, eastern Canada, Europe, western Russia, eastern Russia), as defined based on The Boreal–Arctic Wetland and Lake Dataset (BAWLD) (Olefeldt et al., 2021).

## 2.2. Wetland estimates used as prior fluxes in the inverse modelling

In this study, we utilize the JSBACH model (Reick et al., 2021), the land component of the MPI-ESM (Mauritsen et al., 2019), to estimate bottom-up wetland $CH_4$ emissions. Originally, JSBACH was developed as a lower boundary condition for the atmospheric component of the MPI-ESM; however, it has since been updated to function as a standalone land surface model driven by observed climate data to simulate terrestrial components of the carbon, energy and water cycles. In this study, simulations conducted at T63 resolution (approximately 1.85°, or 185 km) were driven using the CRUJRA2.3 (Harris, 2019) climate data. A multilayer vertical soil profile is implemented as described by Hagemann and Stacke (2015), while features relevant for high-northern latitudes permafrost have been implemented by Ekici et al. (2014). The Richards' equation (Richards, 1931), along with thermal diffusion, governs the vertical distribution of moisture and heat in the soil (Reick et al., 2021). Soil organic carbon (SOC) decomposition is simulated as a first-order decay process that depends on surface air





temperature, water availability, and litter size, following the YASSO model formulation (Tuomi et al., 2011) and its
implementation in JSBACH by Goll et al. (2015).
The wetland area fraction of the grid is determined using TOPMODEL (Beven and Kirkby, 1979), a conceptual rainfall-
runoff model that estimates inundation based on the compound topographic index (CTI). If the inundated fraction of the grid
is non-frozen (depending on the soil temperature), it is considered a $CH_4$-emitting area. The methodology for wetland $CH_4$
production and transport is adopted from Riley et al. (2011), and the details of the TOPMODEL and its implementation for
wetland $CH_4$ within JSBACH are outlined in Kleinen et al. (2020). TOPMODEL assumes a constant exponential decline of
transmissivity with depth, defined as the ratio of the difference between the local sub-grid-scale CTI and the mean grid-cell
CTI to the difference between their corresponding local sub-grid-scale water table and mean grid-cell water table (see equation
(1) in Kleinen et al. (2020)). As water propagates from the surface, it saturates the soil layers based on volumetric moisture
content and field capacity. Starting from the bottom of the soil column, the mean grid-cell water table is located in the first
soil layer where the layer saturation is below the experimentally determined saturation threshold. The sensitivity study
indicates that using CRUJRA (Harris, 2019) as the forcing data, setting the saturation threshold at 7.25, configuring the
exponential decline of transmissivity with depth to 4, and limiting the valid range of CTI to values greater than 5.5 results in
a reasonable estimation of present-day wetland extents.
In JSBACH, carbon enters the soil as litter, both above- and belowground, originating from decomposing vegetation. This
carbon eventually returns to the atmosphere through decomposition processes as $CO_2$ and $CH_4$ emissions. Carbon fixed by
vegetation is allocated to green tissue (leaves, fine roots), wood (stems, branches), and reserve pools (e.g., sugars and starches).
Routine turnover, herbivory, and root exudation transfer carbon into above- and belowground litter pools. Depending on the
plant functional type (PFT), litter carbon is distributed among acid-soluble, water-soluble, ethanol-soluble, and non-soluble
pools, each further divided into above- and belowground fractions, as well as a humus pool. Decomposition rates vary based
on temperature, precipitation, and litter size. Under anoxic conditions (in the inundated fraction of the tile), SOC decomposes
into both $CO_2$ and $CH_4$. The baseline rate of SOC decomposition under anaerobic conditions is reduced compared to aerobic
conditions. Temperature dependency of $CH_4$ production as part of SOC decomposition follows the $Q_{10}$ model with a reference
temperature of 295K (Equation 1). The fraction of $CH_4$ production is capped at 0.5; that is, no more than 50% of carbon can
be converted to $CH_4$. However, the $CH_4:CO_2$ ratio of net emissions to the atmosphere is typically lower than the ratio of gross
production due to oxidation (methanotrophy) and differences in transport pathways. Oxidation, which follows Michaelis-
Menten kinetics (with $Q_{10} = 1.9$, which remained constant throughout the sensitivity tests), converts a portion of $CH_4$ to $CO_2$,
thus increasing $CO_2$ and decreasing $CH_4$ emissions. Transport mechanisms further differentiate the fate of these gases: $CH_4$
can escape via diffusion, plant-mediated transport, or ebullition, whereas $CO_2$ is not released through ebullition. $O_2$ availability
and soil moisture regulate the efficiency of $CH_4$ oxidation. Therefore, the net $CH_4:CO_2$ emission ratio depends on the combined
effects of $CH_4$ production, oxidation, and transport processes. Warmer, oxic conditions tend to reduce the net $CH_4:CO_2$ (due
to stronger aerobic oxidation of $CH_4$), while colder or persistently anoxic, saturated conditions (with ebullition) can increase



the net $CH_4$:$CO_2$ ratio compared to cases with strong oxidation. Equation 1 shows how the $Q_{10}$ law controls the $CH_4$ fraction
($f_{CH_4}$) as a function of soil temperature ($T_{soil}$) and the baseline fraction (baseline $f_{CH_4}$ fraction):

$$f_{CH_4} = f_{CH_4,baseline} \cdot Q_{10}^{(T_{soil}-295)/10K} \qquad \text{Equation 1}$$

To evaluate how sensitive $CH_4$ wetland emission estimates are to key parameters, we conducted nine experiments in which
we varied only the $Q_{10}$ coefficient for $CH_4$ production and the baseline $f_{CH_4}$ fraction (Fig. 2b). Specifically, we tested three
different $Q_{10}$ values ranging from 1.4 to 2.2 and baseline $f_{CH_4}$ fractions from 0.33 to 0.38. These combinations are summarized
in Table 1 and were chosen to identify parameter sets that best align with the observed atmospheric data.

### 2.3. Inverse modeling setup


We used the Jena CarboScope Inversion System (Rödenbeck, 2005) to quantify $CH_4$ emissions between the surface and
the atmosphere globally from 2010 to 2021, with the evaluation and interpretation of fluxes focused on the Arctic-Boreal
region. This is a linear Bayesian framework that infers surface–atmosphere $CH_4$ fluxes based on observed atmospheric mole
fractions. A total of 154 stations were assimilated for the global domain (Fig.1). These $CH_4$ observations were obtained from
several global and regional networks (ICOS RI et al., 2024; Schuldt et al., 2023), with the majority of sites located in the
Northern Hemisphere, including 33 stations within the Arctic–Boreal domain. For tower sites with multiple intake heights
available, we assimilated only data from the highest height in the inversion, and for the continuous data, we use only daytime
measurements. The transport model used in CarboScope is the TM3 global atmospheric tracer model (Heimann and Körner,
2003) and is driven by meteorological inputs from the NCEP reanalysis dataset (Kalnay et al., 1996). Flux inversions were
conducted at a spatial resolution of approximately 3.8° latitude by 5° longitude, with 19 vertical layers and a daily temporal
resolution. To account for model-data mismatch, including the representation error of the measurements within the transport
model, we assigned an uncertainty of 30 ppb. Additionally, to ensure balanced representation across observational sites,
particularly between continuous and sparse time series, we applied a data density weighting scheme, assigning equal influence
to each weekly period, regardless of data frequency (Rödenbeck, 2005).
Prior $CH_4$ flux estimates include five source categories, all of which were optimized: wetlands, other natural sources,
anthropogenic, ocean and fire emissions. The monthly mean emissions from wetlands and fires were obtained from the
JSBACH model (Kleinen et al., 2020), as previously described. Additional natural sources, such as termites and wild animal
emissions taken from JSBACH (Kleinen et al., 2020) and geological emissions from Etiope et al. (2019) were combined as
the "other natural source" category. Emissions from oceans were obtained from Weber et al. (2019) and implemented as a
non-seasonal climatology. Anthropogenic emissions were obtained from the EDGAR inventories database



(https://edgar.jrc.ec.europa.eu) version 8 (IEA et al., 2024) and are provided as monthly global fluxes. This category includes
emissions from agriculture, livestock, waste management, fossil fuel exploitation and other minor anthropogenic sources
except biomass burning.
CH$_4$ chemical loss includes loss due to OH and Cl in the troposphere, as well as OH, Cl, and O($^1$D) in the stratosphere.
For tropospheric OH, we use the monthly three-dimensional OH fields calculated by Spivakovsky et al. (2000), which are
based on observed climatological distributions of OH precursors and scaled to match the observed CH$_3$CCl$_3$ lifetime. The
monthly climatological loss rates of CH$_4$ in the stratosphere due to OH, Cl, and O($^1$D) were derived from a simulation of the
ECHAM5/MESSy1 chemistry transport model (Jöckel et al., 2006). Additionally, tropospheric Cl loss is simulated using a
recent model-derived estimate of tropospheric Cl (Hossaini et al., 2016). The surface sink from upland soils and the ocean was
implemented as a zeroth-order reaction with prescribed reaction rates that occur only in the surface-most model layer. Reaction
rates for the microbial oxidation of atmospheric CH$_4$ in soil were based on the uptake estimates from the LPJ-Bern model
(Spahni et al., 2011).

### 2.4.  Evaluating Bottom-Up Emissions Using Top-Down Constraints

Previous studies have used atmospheric inversion models to evaluate in between different bottom-up estimates which one
best reproduce observed atmospheric CH$_4$ data (e.g., Kim et al., 2011; Miller et al., 2016), providing an effective framework
for model evaluation. In this study, we evaluated the performance of different JSBACH parameterizations by using the CH$_4$
wetland emission outputs from each experiment as wetland prior fluxes in a top-down atmospheric inversion framework. The
inversion then generated posterior fluxes, reflecting the adjustments needed to align the prior emissions with atmospheric CH$_4$
observations. In this study, we used the model adjustment defined as the difference between posterior and prior fluxes,
calculated as the mean monthly and mean annual values across the Arctic–Boreal region from 2010 to 2021. First, we identified
the parameterization resulting in the lowest mean model adjustment across the entire domain. For the monthly analysis, we
first computed the mean monthly prior flux and the mean monthly posterior flux, and then defined the model adjustment as
the difference between these two means. For the annual analysis, we calculated the mean annual prior and posterior fluxes and
again defined the adjustment as their difference. This allowed us to determine which JSBACH configuration provided the best
overall agreement with atmospheric constraints at the pan-regional scale and investigate temporal variability. Next, we
examined spatial variability of the difference between posterior and prior fluxes using different JSBACH parameterizations as
wetland priors. At the grid-cell level, we identified the parameter combination that minimized annual model adjustment,
thereby providing the best match to the top-down atmospheric constraints. To conduct this analysis, an ensemble of posterior
fluxes was calculated based on each CH$_4$ production Q$_{10}$ value from the prior wetland flux. This approach was supported by
the observation that CH$_4$ production Q$_{10}$ significantly influenced CH$_4$ emission estimates compared to the baseline
$f_{CH_4}$ fraction. Additionally, posterior fluxes from priors with different baseline $f_{CH_4}$ fraction scenarios remained highly similar
for a given Q$_{10}$ value. As a result, maps were made by calculating the absolute difference between the posterior ensemble of

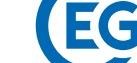

240 the respectively $Q_{10}$ value and prior $CH_4$ fluxes for each experiment at each grid-cell. Then, the annual mean adjustment was

241 calculated and we identified the parameterization that resulted in the smallest adjustment at each grid-cell. In summary, each

242 grid-cell shows the experiment that best matched the atmospheric $CH_4$ observations.

243 **3. Results and Discussion**

244 **3.1 Sensitivity of JSBACH $CH_4$ wetland emission estimates to $CH_4$ production $Q_{10}$ and baseline $f_{CH_4}$ fraction in**

245 **Arctic–Boreal region**

246 Table 1 summarizes the experiments and parameters combinations that have been tested in the JSBACH model and used

247 as a wetland prior in the atmospheric inversions. Across the Arctic-Boreal region, our nine experiments produced annual mean

248 $CH_4$ wetland estimates ranging from 13.8 to 33.5 $TgCH_4$ $y^{-1}$. These estimates are consistent with previously published bottom-

249 up estimates of ~15-50 $TgCH_4$ $y^{-1}$ per year, with most studies reporting mean values near 20-25 $TgCH_4$ $y^{-1}$ (Christensen et al.,

250 1996; Ying et al., 2025; Yuan et al., 2024; Zhang et al., 2025). It should be noted that these studies consider different spatial

251 domains and time periods. The estimates obtained using a $Q_{10}$ value of 1.8 align most closely with this published range among

252 our experiments.


254 Table 1. Summary of JSBACH wetland $CH_4$ estimates used as prior fluxes in the inversions and posterior fluxes estimates for

255 each respective model run.

| Experiment | JSBACH parameterization | | Arctic-Boreal annual mean $CH_4$ emission ($TgCH_4$ $y^{-1}$)* | | |
| :---: | :---: | :---: | :---: | :---: | :---: |
| | Baseline $f_{CH_4}$ fraction | $Q_{10}$ model | JSBACH estimates (prior) | Posterior estimates | Mean model adjustment |
| B1_low | 0.33 | 1.4 | 31.7 ± 1.1 | 25.0 ± 1.4 | -6.7 |
| B1_mid | 0.33 | 1.8 | 20.0 ± 0.7 | 22.9 ± 1.1 | 2.9 |
| B1_high | 0.33 | 2.2 | 14.6 ± 0.5 | 21.2 ± 0.9 | 6.6 |
| B2_low | 0.35 | 1.4 | 29.7 ± 0.9 | 24.8 ± 1.5 | -5.0 |
| B2_mid | 0.35 | 1.8 | 18.9 ± 0.6 | 22.7 ± 1.1 | 3.8 |
| B2_high | 0.35 | 2.2 | 13.8 ± 0.5 | 20.9 ± 0.9 | 7.1 |
| B3_low | 0.38 | 1.4 | 33.5 ± 1.0 | 25.2 ± 1.6 | -8.2 |
| B3_mid | 0.38 | 1.8 | 21.3 ± 0.7 | 23.3 ± 1.2 | 2.0 |
| B3_high | 0.38 | 2.2 | 15.5 ± 0.5 | 21.6 ± 1.0 | 6.1 |

256 *The annual mean between 2010 and 2021, with the standard deviation representing interannual variability.




Emissions peaked during the summer months (July-August), with a mean emission ranging from 6.8 to 14.1 TgCH$_4$ y$^{-1}$
(Fig. 2b). These larger emissions were followed by spring (May-June; range of 3.5-7.8 TgCH$_4$ y$^{-1}$), autumn (September-
October; range of 2.8-7.7 TgCH$_4$ y$^{-1}$), and winter with the lower emissions (November-April; range of 0.4-1.5 TgCH$_4$ y$^{-1}$). The
timing of the peak in wetland emissions aligns with previous bottom-up estimates (Ying et al., 2025). At the sub-regional scale,
emissions showed substantial spatial variability (Fig. 2c). The highest annual mean fluxes were found in western Russia (3.4-
8.7 TgCH$_4$ y$^{-1}$, depending on the parameter set), followed by eastern Canada (3.4-8.2 TgCH$_4$ y$^{-1}$), eastern Russia (3.1-7.2
TgCH$_4$ y$^{-1}$), western Canada (1.8-4.4 TgCH$_4$ y$^{-1}$), Europe (1.5-3.4 TgCH$_4$ y$^{-1}$), and Alaska (0.5-1.6 TgCH$_4$ y$^{-1}$).
In general, increasing the baseline value of the $f_{CH_4}$ fraction from 0.33 to 0.38 increases CH$_4$ production. However, an
increase in the CH$_4$ production Q$_{10}$ parameter decreases CH$_4$ production for temperatures below 295 K (the reference
temperature) and increases it for temperatures higher than 295 K. This means that increasing Q$_{10}$ values from 1.4 to 2.2 reduces
wetland CH$_4$ emissions in the comparatively cold Arctic region (Table 1 and Fig. 2). The sensitivity of wetland CH$_4$ to the Q$_{10}$
temperature response and the baseline $f_{CH_4}$ fraction is evident when comparing seasonal cycles over the Arctic-Boreal domain
(Fig. 2b). For example, contrasting the simulations with baseline $f_{CH_4}$ fraction equaling 0.33 and varying CH$_4$ production Q$_{10}$
values (from 1.4 to 2.2), shows that increasing Q$_{10}$ significantly reduces annual wetland mean CH$_4$ emission in this region by
~ 54% (~17 TgCH$_4$ y$^{-1}$). This reduction is not uniform throughout the year. Although winter emissions are relatively low,
increasing Q$_{10}$ from 1.4 to 2.2 results in a ~72% decrease compared to a ~50-59% decrease during the summer, spring and fall.
Similarly, the influence of the baseline $f_{CH_4}$ fraction can be observed by keeping Q$_{10}$ constant, for example at 1.4, and varying
the baseline $f_{CH_4}$ fraction from 0.33 to 0.38. This increase leads to an increase of up to 6% in the annual wetland CH$_4$ emissions
for the region. In general, our parameter sensitivity tests show that CH$_4$ production Q$_{10}$ has a stronger effect on emission
variability than the baseline $f_{CH_4}$ fraction. These wetland CH$_4$ emission estimates with different parameterizations were
subsequently integrated into the Jena CarboScope atmospheric inversion framework as wetland prior fluxes to determine the
combination that closest align with atmospheric CH$_4$ observations, which means those requiring the minimum adjustment to
fluxes from prior to posterior.



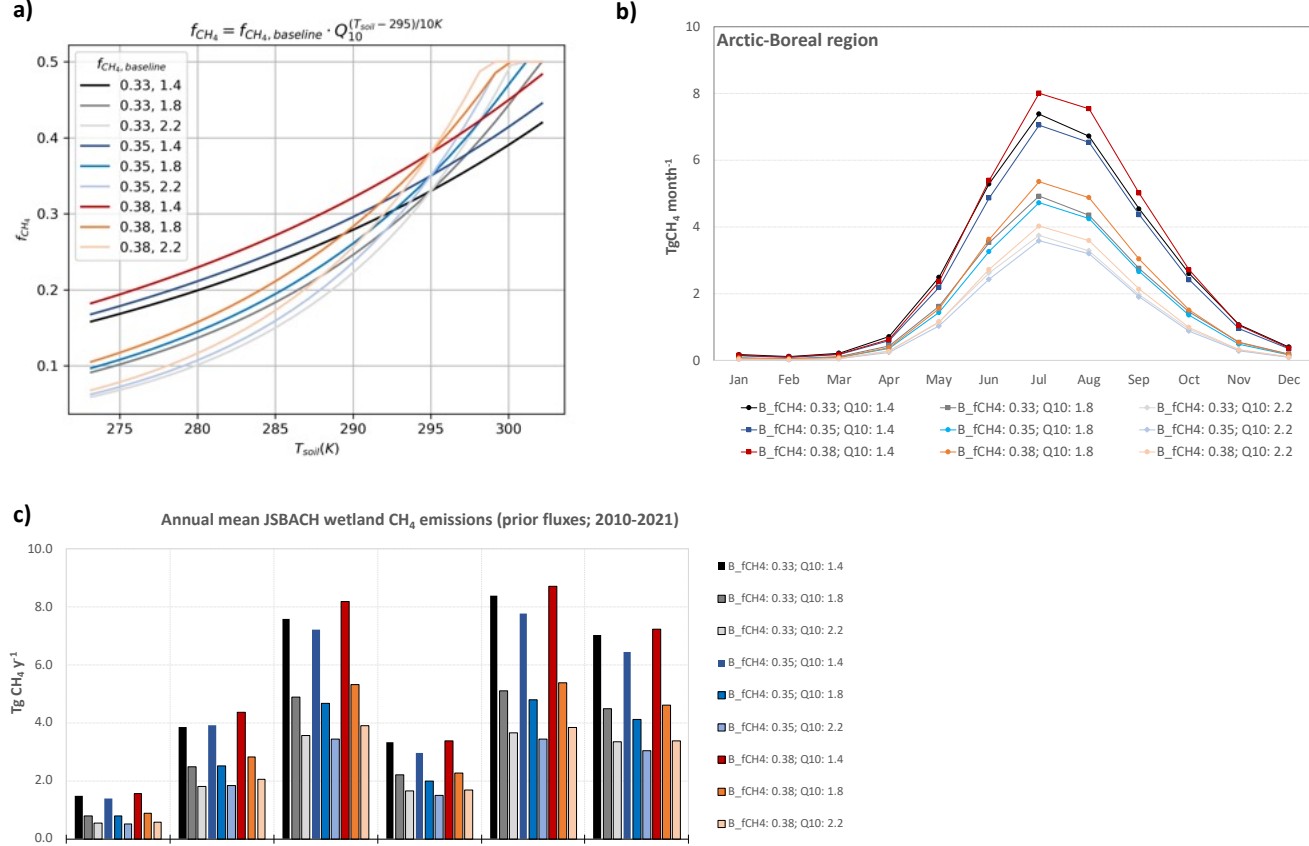

**Figure 2.** a) Sensitivity of $f_{CH_4}$ production fraction to the chosen range of input parameters for this study. The y-axis represents the fraction of anaerobic carbon mineralization allocated to $CH_4$ production, calculated using the equation displayed at the top of the panel and in Equation 1. In the legend, the first number denotes the $f_{CH_4}$ baseline fraction and the second number denotes the $CH_4$ production $Q_{10}$ value. b) Mean seasonal cycle of Arctic-Boreal wetland $CH_4$ emissions for each experiment used in the inversion as the wetland prior flux. c) Annual mean wetland fluxes from each experiment estimated by JSBACH model.

### 3.2 Evaluation of JSBACH CH₄ Fluxes Using Inverse Modeling

Our nine inverse model estimates produce an annual mean total emission (i.e. including natural and anthropogenic sources) for the Arctic-Boreal region ranging from 44.2 to 47.1 $TgCH_4$ $y^{-1}$, with wetland emissions being the main $CH_4$ source to the atmosphere. Depending on the parameter set in prior flux setup by JSBACH, the annual mean wetland emission ranges from 20.9 to 25.0 $TgCH_4$ $y^{-1}$ (47-54% of total emissions). The largest posterior wetland $CH_4$ emissions were estimated for western Russia (range of 6.9-8.4 $TgCH_4$ $y^{-1}$, depending on the parameter set), followed by eastern Russia (range of 6.0-7.5 $TgCH_4$ $y^{-1}$), eastern Canada (range of 4.3-4.9 $TgCH_4$ $y^{-1}$), western Canada (range of 1.7-1.8 $TgCH_4$ $y^{-1}$), Alaska (range of 1.0-2.0 $TgCH_4$ $y^{-1}$) and Europe (range of 0.7-0.8 $TgCH_4$ $y^{-1}$)

At the pan-Arctic scale, posterior wetland fluxes are higher than prior fluxes in the experiments using $CH_4$ production $Q_{10}$ values of 1.8 (8-22% higher than prior) and 2.2 (37-54% higher), see Table 1 and Fig. 3a. This suggests that these prior





estimates underestimate $CH_4$ emissions in the Arctic–Boreal region relative to the observation-constrained posterior fluxes.
However, prior fluxes estimated using a $Q_{10}$ value of 1.4 are higher than posterior fluxes (16-25% higher than posterior),
indicating overestimation of $CH_4$ emissions in this case. When comparing the model adjustment for the three experiments
(varying only the $Q_{10}$ parameters), the prior flux using $Q_{10}$ values of 1.8 produces the best agreement between prior and
posterior flux budgets, meaning that a minimum adjustment in the inverse model optimization is required when considering
annual mean emissions in the entire Arctic-Boreal region. Additionally, when comparing the different baseline $f_{CH_4}$ fractions
(using the $Q_{10}$ value with the best fit: 1.8), the minimum adjustment in the inverse model optimization is required for the prior
flux with the largest baseline $f_{CH_4}$ fraction (0.38), with posterior flux being 8% (2.0 $TgCH_4$ $y^{-1}$) higher than the prior.

307        Our posterior estimates of $CH_4$ emissions from wetlands are similar to previous Arctic-Boreal estimates. Using a process-

oriented ecosystem model, Christensen et al. (1996) estimated a total $CH_4$ emissions from northern wetlands and tundra (>
50°N) to be $20 \pm 13$ $TgCH_4$ $y^{-1}$. Yuan et al. (2024) reported a mean annual emission of $20.3 \pm 0.9$ $TgCH_4$ $y^{-1}$ from boreal-Arctic
wetland based on upscaled flux observations for the period 2002-2021. The Global Carbon Project estimated a mean annual
wetland (including inland freshwaters) $CH_4$ emission for regions north of 60°N at 24 (9-53) $TgCH_4$ $y^{-1}$, while top-down
approaches resulted in a lower estimate of 9 (7-17) $TgCH_4$ $y^{-1}$ for the same region (Saunois et al., 2025). Recently, Ying et al.
(2025) estimated an annual mean $CH_4$ emissions from vegetated wetlands north of 45°N during 2016-2022 at
$22.8 \pm 2.4$ $TgCH_4$ $y^{-1}$, ranging from $15.7 \pm 1.8$ $TgCH_4$ $y^{-1}$ to $51.6 \pm 2.2$ $TgCH_4$ $y^{-1}$, depending on the wetland dataset used in the
machine-learning-based upscaling approach. Although our posterior estimates are within the range of previous Arctic-Boreal
estimates, direct comparisons are difficult because of differences in the study period, methodological approach, and
inconsistent or unclear definitions of the spatial domain.

**3.3 Seasonal variability in optimum $CH_4$ production $Q_{10}$ settings**

320        Before analyzing regional differences in optimum $CH_4$ production $Q_{10}$ settings, we first focused on a clear seasonal

pattern in the adjustments between prior and posterior $CH_4$ emissions, which showed a peak of changes occurring during
summer. We therefore assessed whether the $Q_{10}$ value resulting in the minimum adjustment remained constant throughout the
year or varied by season. At a pan-Arctic scale, seasonal variations were evident: estimates using $CH_4$ production $Q_{10}$ equaling
1.8 aligned better with atmospheric observations in spring and fall but substantially underestimated summer emissions (Fig.
3b). In contrast, estimates using a $Q_{10}$ of 1.4 best agreed well with the atmospheric observation during summer, reducing the
discrepancy between top-down and bottom-up estimates during the growing season, but strongly overestimating emissions in
spring and fall (Fig. 3b). This pattern is primarily driven by wetlands in Russia. Bergman et al. (2000) found temporal variation
in $Q_{10}$ at peatland sites, suggesting that factors such as the availability of easily degradable compounds (e.g., root exudates)
and the activity of anaerobic microbial biomass influence $CH_4$ production rates alongside temperature.





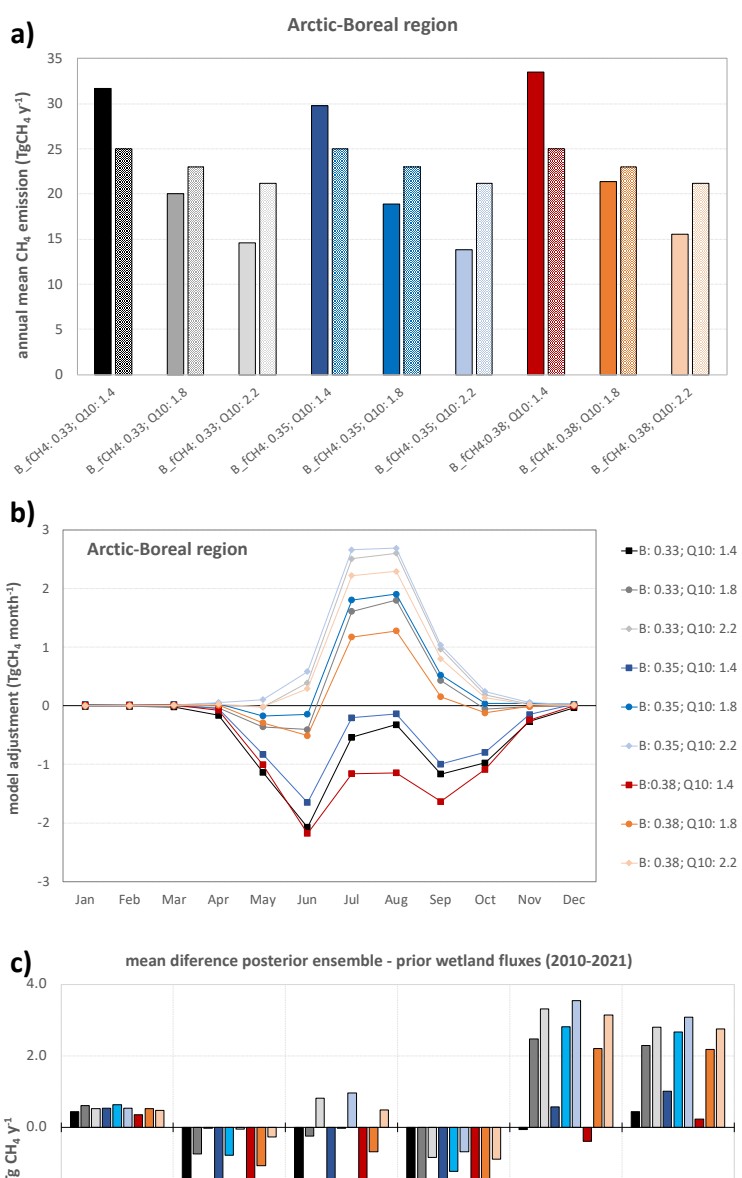

Figure 3. a) Annual mean CH₄ emissions (prior: full color bars; posterior: light color bars) for the entire Arctic-Boreal region using different values of $Q_{10}$ parameter and baseline $f_{CH_4}$ fraction in JSBACH wetland emissions. b) adjustment of prior fluxes at monthly timesteps for the same model configurations as used in (a). c) annual mean model adjustment (posterior minus prior flux) for each one of the sub-regions. Positive values indicate regions where prior estimates underestimated emissions compared with posterior estimates, while negative values represent areas where prior emissions overestimate CH₄ emissions compared with the posterior estimates.






**3.4. Spatial patterns of best-fit model results based on posterior fluxes**
$CH_4$ emissions exhibited spatial variability, and model adjustments were not uniform across the domain. This suggests
that the optimal parameterization varies by region and seasons (as discussed in Section 3.3). In some areas, $Q_{10}$ values of 1.4
or 2.2 resulted in minimal adjustments (Fig. 3c), outperforming the model using a $Q_{10}$ equaling 1.8 that was shown to work
best as an average setting across the entire domain. To better evaluate this variability and explore ways to reduce uncertainty
in specific regions, we assessed the best parameterization fit with observations at the per grid-cell level (Fig. 4).
In our first analysis, we evaluated the spatial best fit model by keeping the baseline constant at a value of 0.35 and varying
the $CH_4$ production $Q_{10}$ values (Fig. 4a). This spatial analysis showed that, in general, in regions with large wetland areas and
high annual $CH_4$ emissions (for example the Western Siberian Lowlands) a $Q_{10}$ value of 1.4 resulted in the smallest model
adjustment. As an increase in the $Q_{10}$ parameter decreases $CH_4$ production for temperatures below 295 K, a higher $Q_{10}$ value
in these regions results in an underestimation of emissions. In contrast, regions such as Europe and northern Canada showed,
in general, minimum model adjustments with a $Q_{10}$ value of 2.2, suggesting that lower $Q_{10}$ value would overestimate wetland
$CH_4$ emissions in these regions. Interestingly, we observed adjustments with different signs in eastern Canada depending on
the parameterization. For example, positive adjustments were associated with $Q_{10}$ value of 2.2, as the prior emissions were
underestimated compared with the estimated flux inferred from atmospheric observations. Additionally, we analyzed the effect
of varying baseline flux values while keeping $Q_{10}$ constant as 1.8, which showed that in high-emission areas, for example the
Western Siberian Lowlands, in general a larger baseline flux value led to the smallest model adjustments (Fig. 4b). When
considering the model adjustment for all sensitivity tests (varying both $CH_4$ production $Q_{10}$ and baseline $f_{CH_4}$ fraction) as shown
in Fig. 4c, we also found a consistent pattern that confirmed the above findings varying only single parameters: the combination
of higher baseline fluxes and lower $Q_{10}$ value ($Q_{10} = 1.4$) best captured $CH_4$ dynamics in $CH_4$ hotspots, as the Western Siberia
Lowlands.
The wide range of reported incubation-based $Q_{10}$ values for $CH_4$ production in Arctic and northern wetlands depending
on the site, substrate, and season, shows that the temperature sensitivity of $CH_4$ production varies considerably across
environments (Bergman et al., 2000; Roy Chowdhury et al., 2015; Treat et al., 2015). This variability, which could be driven
by factors such as vegetation type, organic matter quality, and microbial activity, emphasizes the necessity of models to account
for spatial differences in process rates. For example, one synthesis study reported a mean $Q_{10}$ value of 1.18 for $CH_4$ production
under Arctic soil conditions (Treat et al., 2015). Roy Chowdhury et al. (2015) used anoxic laboratory incubations of active
layer and permafrost samples from the Barrow Environmental Observatory in Alaska and reported a range of $Q_{10}$ values from
1.8 to 22. Lupascu et al. (2012) reported that $Q_{10}$ values describing the $CH_4$ production response of peat to a 10 °C temperature
change ranged from 1.9 to 3.5 in sedge sites and from 2.4 to 5.8 in *Sphagnum* mire sites, and suggested that using spatially
variable $CH_4$ production $Q_{10}$ values could improve the accuracy of $CH_4$ flux modeling in northern wetlands. Furthermore,
Bergman et al. (2000) found that the seasonal average $Q_{10}$ values ranged from approximately 4.6 to 9.2 depending on the plant
community of the various peat types. Here, our intent is not to directly compare our results with reported incubation-based



values, since our adjustments in the $CH_4$ production $Q_{10}$ refer to the $Q_{10}$ of the $CH_4{:}CO_2$ production ratio, as represented in the
model, and could not directly be comparable with $CH_4$ production $Q_{10}$ from the literature review. In JSBACH, the $Q_{10}$ applied
to $CH_4$ production controls the fraction of $CH_4$ generated, but the surface emission ratio may still be lower due to oxidation
and transport pathways. Together, these examples highlight that $CH_4$ production are strongly temperature dependent, and that
the degree of this dependency can differ across regions and time periods. However, most current models cannot fully capture
the influence of these factors due to structural limitations or a lack of detailed input data that is both spatially and temporally
resolved. Consequently, these environmental drivers are often oversimplified or overlooked. Adjusting the $CH_4$ production
$Q_{10}$ values, as we do here, offers a useful initial approach, but it should not be seen as a long-term solution. Ideally, future
model and data developments will enable $CH_4$ production $Q_{10}$ values to adjust dynamically in response to underlying
biophysical conditions, such as shifts in vegetation or organic matter characteristics. This will allow models to operate with a
more generalizable formulation that still captures observed heterogeneity. Although our model experiments identified a single
$CH_4$ production $Q_{10}$ value that best agrees with observations at the pan-Arctic scale, they also showed that $CH_4$ emissions and
model adjustments vary regionally. Some areas showed a substantial response to different $Q_{10}$ values, which further
demonstrates that an approach using a single parameter value is not sufficient.




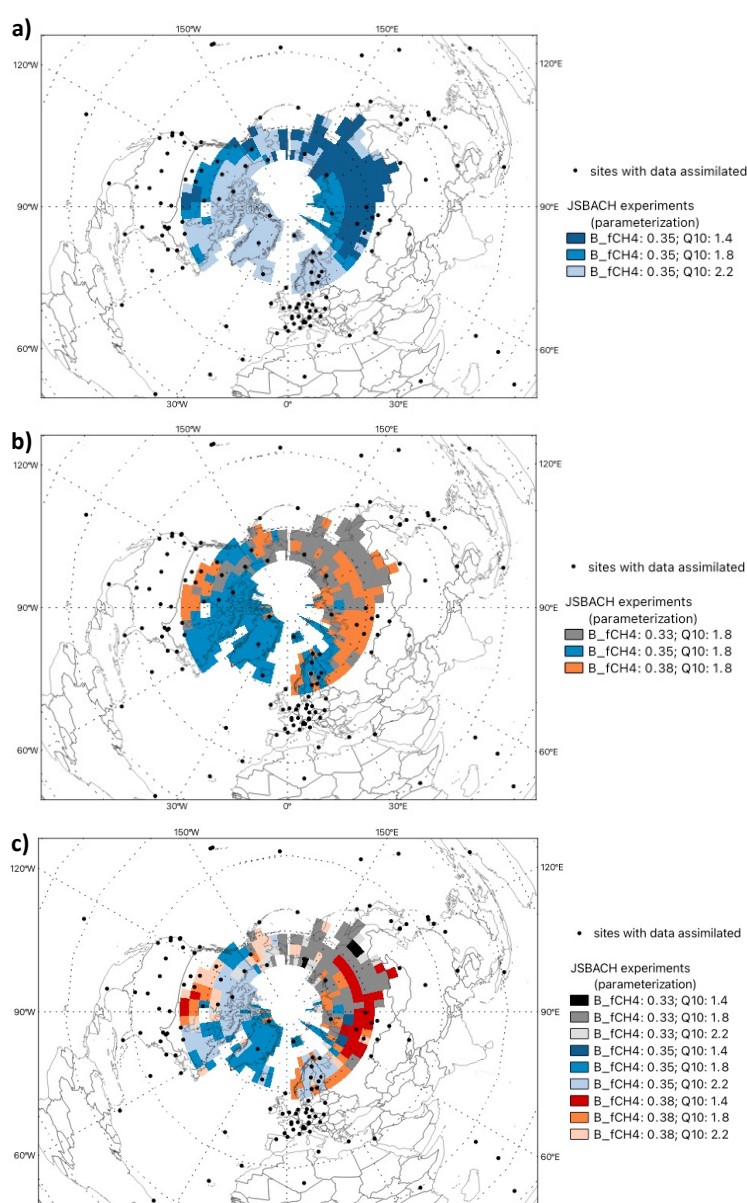

Figure 4. Map of the prior flux setting leading to minimum model adjustment (posterior minus prior fluxes) for the annual mean fluxes at each grid-cell for the Arctic-Boreal region varying the (a) $CH_4$ production $Q_{10}$ parameter only, (b) baseline $f_{CH_4}$ fraction only and (c) both $Q_{10}$ parameter and baseline $f_{CH_4}$ fraction.





## 4. Conclusions


Overall, our parameter sensitivity tests of bottom-up wetland emissions indicate that $CH_4$ production $Q_{10}$ has a stronger
effect on emission variability than the baseline $f_{CH_4}$ fraction. Our bottom-up estimates showed that increasing $CH_4$ production
$Q_{10}$ from 1.4 to 2.2 decreased the annual mean wetland $CH_4$ emission in the Arctic–Boreal region by half. In addition, our
analysis shows that a single $Q_{10}$ value cannot capture the complexity of $CH_4$ emission dynamics across the Arctic-Boreal
region. $CH_4$ production $Q_{10}$ values of 1.8 and 2.2 underestimate hotspot emissions, mainly during summer. In contrast, a $Q_{10}$
value of 1.4 overestimates emissions in regions with lower annual mean wetland emissions, such as e.g., Europe and northern
Canada. Furthermore, a baseline $f_{CH_4}$ fraction value of 0.38 led to the smallest model adjustments in $CH_4$ hotspots. These
findings emphasize the importance of selecting appropriate parameterizations to accurately represent wetland emissions,
especially in regions with substantial $CH_4$ release. Future models should incorporate dynamic, data-driven adjustments to
reflect underlying environmental controls more accurately. If a varying $CH_4$ production $Q_{10}$ value approach is not feasible for
this region due to computational cost or model setup constraints, using a $Q_{10}$ value of 1.8 provides the more similar $CH_4$
emission estimates compared to the atmospheric data across the entire Arctic-Boreal region.
Our analysis shows that atmospheric inverse modeling is a useful tool for evaluating and guiding process-model
parameterizations when estimating wetland $CH_4$ emissions. However, it is important to note the limitations of the top-down
approach. Top-down estimates rely heavily on the spatial and temporal distribution of atmospheric observations incorporated
into the model. Regions with limited data or gaps, such as eastern Russia, can limit the ability to accurately identify emission
sources and increase dependence on prior estimates. Global atmospheric inversions often operate at coarser spatial resolutions
than the process models, and emission variability, including hotspot emissions, reducing the ability to estimate local scale
process. At the grid-cell scale, assimilating only atmospheric $CH_4$ observations that is a result of total emissions (the balance
between all sources and sinks) does not differentiate the overlapping source sectors in a grid-cell. However, differences in the
spatial patterns and seasonality of emissions can be constrained by atmospheric $CH_4$ observations in inversions that solve for
different sources categories (Saunois et al., 2025). Furthermore, errors in atmospheric transport model can propagate into
emission estimates (Houweling et al., 1999; Locatelli et al., 2013; Schuh et al., 2019). Despite these limitations, our approach
demonstrated a strong potential to help reduce the discrepancy between bottom-up and top-down estimates, therefore
improving the accuracy of wetland $CH_4$ emission estimates.

## 5. Authors contributions


LSB, MG, GG, VB designed the methodology. LSB wrote the first version of the manuscript and performed analysis and $CH_4$
inversions. GG performed and provided the JSBACH simulations. CR provided guidance and technical support for the inverse
modelling. CB provided additional input on the discussion of results. All authors contributed with analysis and text. MG
supervised and acquired funding.





## 6. Competing interests

The authors declare that they have no conflict of interest.

## 7. Acknowledgements

The authors were funded by the European Research Council (ERC synergy project Q-Arctic, grant agreement no. 951288), the German Federal Ministry of Research, Technology and Space (MOMENT project, support code 03F0931G), and the AMPAC-net initiative (European Space Agency, grant no. 4000137912/22/I-DT). We would like to thank all Principal Investigators and supporting staff for setting up and maintaining observation sites around the world, particularly in the Arctic, and for making the data available through different databases. The authors would also like to thank Santiago Botía at MPI-BGC/BSI for his valuable comments and suggestions, which helped us to improve this manuscript. The authors would like to acknowledge the contributions of Tonatiuh Nunez Ramirez, who designed the $CH_4$ chemistry model for CarboScope inversion system used in this work. Parts of the text was language-edited for grammatical correctness using DeepL. The authors have reviewed and verified the content as needed and take full responsibility for it.

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
