# Peer review of "A top-down evaluation of bottom-up estimates to reduce uncertainty in methane emissions from Arctic wetlands"

_EGUsphere, 2025_

## Referee Comment (RC1)

**A top-down evaluation of bottom-up estimates to reduce uncertainty in methane emissions from Arctic wetlands**

Luana S. Basso, Goran Georgievski, Victor Brovkin, Christian Beer, Christian Rödenbeck, and Mathias Göckede

https://doi.org/10.5194/egusphere-2025-4467

**General comments:**

This study is focused on wetlands in the Arctic boreal region. The authors alter the parameters  $f_{\text{CH4}}$  (fraction of baseline methane) and  $Q_{10}$  in JSBACH to produce a total of 9 forward simulations. They compare these to eachother, then use them as priors in atmospheric inversions from 2010 to 2021. The inversion that yields the smallest difference between prior and posterior fluxes is interpreted as indicating the JSBACH configuration that best matches the atmospheric observations, evaluated at annual, monthly, and regional scales. The results show that different combinations of  $Q_{10}$  and  $f_{\text{CH4}}$  perform best across different seasons and subregions, highlighting the need for regionally and seasonally varying process model parameters. Overall, the study provides a valuable proof of concept demonstrating how atmospheric observations can help guide the optimization of process-model parameters.

I believe the paper makes a meaningful contribution to the integration of bottom-up and top-down approaches. However, several sections require significant improvement to more clearly convey the key messages and strengthen the methodological transparency. I recommend publication after consideration of the comments below.

**Improvement of general paper structure and text.**

The introduction provides a thorough overview of methane biogeochemistry and modeling approaches, but it currently reads more like a literature review than a focused lead-in to the study. The research gap and specific objectives only become clear after several pages. I suggest condensing the detailed process descriptions (e.g., lines 43–78) and emphasizing earlier why reconciling bottom-up and top-down CH4 estimates is scientifically important and what this study contributes. A shorter, more focused introduction would make the study's novelty clearer to readers.

**Some specifics:**

• There is not enough emphasis on why improving process models in the Arctic specifically is important. This context could appear near the start. Highlighting the magnitude or

consequences of Arctic top-down vs. bottom-up discrepancies would strengthen the motivation.

- The discussions of bottom-up and top-down approaches are disjointed, even though their integration is central to the paper.
- You could note that process models are often evaluated against flux or site-level data, but less commonly constrained by atmospheric observations. That contrast nicely sets up the study's novelty.
- The research question appears midway through, before introducing top-down methods. The introduction should build logically from the knowledge gap to the research question, with the final paragraph outlining how the study addresses it.

Similar to the introduction, Section 2.2 is longer than needed and includes detail that is not directly relevant to the study's research question or required to understand the methodology. Much of this description could be replaced with references to existing JSBACH publications or model documentation. For example, lines 146–158 provide an in-depth explanation of inundation and hydrology processes, which are not examined in the subsequent analysis. Parts of the following paragraph (lines 159–178), could be summarized briefly, cited, or moved to a supplement or appendix. As a reader with an inversion rather than process modelling background, I found this paragraph (lines 159-178), difficult to follow. It aims to introduce the relationship between fCH4 and the CH4:CO2. I would suggest revising this section and perhaps including a mathematical relationship between fCH4 and CH4:CO2.

**Discussion and conclusion:**

As you state, this work provides a valuable initial approach but not a long-term solution -future models will need to allow dynamic changes in  $Q_{10}$  to reflect varying environmental conditions. I think this study is timely and also naturally leads into a broader discussion of methane data assimilation techniques, which aim to optimise process-model parameters using observational constraints. For example, Montiel et al. (2025) conduct inversions with a process model (LPJ-GUESS) that has been optimised using eddy covariance fluxes and compare these with simulations using unoptimised parameters, drawing conclusions about the potential of data assimilation. Bernard et al. (2025) also discusses this direction. However, these studies do not yet address regional or seasonal optimization, which your results highlight as particularly important. I think a discussion on this would support the position of your work.

In the conclusions section, almost the entire second paragraph focuses on the general limitations of using top-down inversions. This material would fit better in the discussion, where reflection on the limitations of your own inversion setup should be included. Shifting it there would help keep the conclusions concise and focused on the study's main findings.

**Data and code availability**

I notice there is no data or code availability section. Please could this be included for transparency? Could you please properly acknowledge the data providers?

**Specific comments:**

Line 36: bottom-up methods also include inventories

Line 98: "used a prior" or "used as priors"

Line 102: Do you mean the combined wetlands and inland freshwater sources?

Line 114: Can you make it clear here or somewhere before how fch4 relates to CO2:CH4.

Section 2.1 Please could you include a list of sites and site years as a table. Potentially in an Appendix?

Figure 1. Why is there a gap between Alaska and Eastern Russia regions?

Line 140: What is the resolution of the climate data?

Line 155: As I mentioned above, I don't think that this level of description is necessary, however here 'the sensitivity study' is mentioned, but I am not sure what that is? Is that in this paper?

Line 184: Please can you justify why you chose these values and their limits?

Line 188: Since this is a linear Bayesian system, a short description of the cost function or posterior solution would be useful.

Line 196: Are you optimizing by grid cell? By region? Are you running TM3 daily and optimizing daily?

Line 196: What is the resolution of the meteorology?

Line 197: This spatial resolution seems quite coarse. Perhaps this could also be discussed in the limitations.

Line 199: So, is the model data mismatch the same for every site? Are your covariance matrices diagonal, or do you take into account spatial/temporal correlations?

Line 200: Please could you provide a short description of the weighting scheme?

Line 202: I think I understand from later in the paper that the emissions are optimized together and not separately (i.e. one scaling for all sources). Please could you make that clear here? Perhaps this could also be discussed in the limitation section.

Lines 202: Please include information on the fire emissions prior! And also, you don't include an inland water/freshwater prior. Please mention this.

Line 225: Taking into account uncertainties in the model/transport etc.

Section 2.4: Could this section be broken into multiple paragraphs for readability? Perhaps one for each top-down result (pan-Arctic, seasonal, regional).

Figure 2: Figure 2a is a bit fuzzy. Could this be exported as a vector file? Maybe in a and b you could use colors/patterns that nicely show how the results are grouped more by Q10 than they are by fch4. I think the B labelling is not necessary, you also use this in Table 1, but B1\_low etc. is not used in the text anywhere.

Figure 3. In c, it's quite difficult to see Alaska, perhaps you could plot this as percentage difference? There is a small typo in the title "difference"

Figure 4. I think these results are really important! Please could you 'zoom-in' on the Arctic-boreal region? Perhaps you could highlight in the legend the experiment that was the best on a pan-Arctic scale, which is clearly not the best regionally.

**References**

Monteil, G., Theanutti Kallingal, J. & Scholze, M. CH4 emissions from Northern Europe wetlands: compared data assimilation approaches. *Atmospheric Chemistry and Physics* **25**, 14251–14277 (2025).

Bernard, J. et al. Satellite-based modeling of wetland methane emissions on a global scale (SatWetCH4 1.0). Geoscientific Model Development **18**, 863–883 (2025).

---

## Referee Comment (RC2)

The authors present a study on the evaluation of nine different scenarios of methane (CH4) wetland emissions in the Arctic, obtained by varying two parameters of the JSBACH land surface model: Q10 and fCH4. These scenarios were evaluated using an inverse modeling approach by analyzing the necessary adjustment of the model between prior and posterior CH4 fluxes. The authors found that a Q10 value of 1.8 generally produced the best prior emission scenario in the pan-Arctic region. However, at a regional scale, the optimal parameter set-ups varied, highlighting the importance of using specific parameters of different regions.

In my opinion, this study has been well prepared and carefully thought out, and no major adjustments are required. However, there are three aspects of the description of the study set-up and manuscript structure that could be improved:

1. I would suggest revising the introduction and condensing the information provided slightly. While it is interesting to read, I think it could be shortened slightly to focus more on the research questions being discussed.

2. I would also suggest describing the observation network used more extensively, and properly acknowledging the institutions that provided the observations. In my opinion, the terms currently used in the study, such as "different databases" or "several global and regional networks", are insufficient. Additionally, the limitations of the in situ network, such as the lack of observation sites in Siberia, should be discussed earlier in the manuscript, as these can have a significant impact on posterior CH4 emissions.

3. Please provide a more thorough description of the inversion set-up in section 2.3, as several aspects have not been sufficiently described so far. For example, how did you define the transport error, and which uncertainties were used for the prior emissions? Did you optimize the total CH4 fluxes, or were the fluxes optimized by source category? This is unclear from the description. How were the initial concentrations defined? You could also potentially include one or two more sentences describing the transport model used.

**Specific comments**

P1, L17:
Would it be possible to already give a short definition of what the Q10 value indicates in the abstract?

P4, L123 and P5, Fig.1:
I would consider renaming the "Europe" region "Europe and Greenland", given that Greenland constitutes a substantial part of this region (even though it belongs to Denmark, it is politically independent and not on the European continent).

P4, L127:
It would be good to mention here that you are using in situ data for the inversion, since "data coverage" could also include satellite data.

P6, L168-169:
Out of curiosity, is the capped fraction of 0.5 a default of the model or a setting of your choice?

P7, L184:
How did you define the ranges of Q10 and fCH4? Are these based on experience and/or other studies?

P7 and P8, Section 2.4
Would it be possible to summarize the described calculations for the evaluation in one or multiple equations?

P11, L290-L 196:
Did the inversion optimize the total CH4 emission or was each emission category optimized separately? In the firs case, how were the wetland emission obtained? (See also general comment 3)

P13, Figure 3a:
So these are the total CH4 emissions from all sources using mean values of all 9 emission scenarios? "using different values of Q10 parameter and baseline fCh4 fraction" is a bit vague and could indicate, that only specific scenarios were used. Also it could be beneficial to plot a pattern in either the prior or the posterior bars since the color difference not always clear (e.g. [https://matplotlib.org/stable/gallery/shapes_and_collections/hatch_style_reference.html](https://matplotlib.org/stable/gallery/shapes_and_collections/hatch_style_reference.html))

P14, L353-L354:
I think it could be helpful to also provide exemplary maps of the prior fluxes (not just the model adjustment) to better visualize expressions such as "which showed that in high-emission areas, for example the Western Siberian Lowlands…"

**Technical corrections**

P3, L84-L85:
Please check grammar, e.g. "One big research question now is how high the Q10 value should be for this temperature dependency of the CH4:CO2 production ratio. In order to answer this question, we employ…"

P8, L221-L222
Please check grammar, e.g. "Previous studies have used atmospheric inversion models to evaluate different bottom-up estimates and determine which best reproduces observed atmospheric $CH_4$ data…"

P8, L239:
Better: maps were created

P12, L325:

Please check grammar: "…best agreed well with…"

---

## Author Comment (AC1)

**Referee 1**

Thanks for your very helpful comments and suggestions. Please find below our answers for each general and technical comment.

**General comments:**

*This study is focused on wetlands in the Arctic boreal region. The authors alter the parameters fCH4 (fraction of baseline methane) and Q10 in JSBACH to produce a total of 9 forward simulations. They compare these to each other, then use them as priors in atmospheric inversions from 2010 to 2021. The inversion that yields the smallest difference between prior and posterior fluxes is interpreted as indicating the JSBACH configuration that best matches the atmospheric observations, evaluated at annual, monthly, and regional scales. The results show that different combinations of Q10 and fCH4 perform best across different seasons and subregions, highlighting the need for regionally and seasonally varying process model parameters. Overall, the study provides a valuable proof of concept demonstrating how atmospheric observations can help guide the optimization of process-model parameters.*

*I believe the paper makes a meaningful contribution to the integration of bottom-up and top-down approaches. However, several sections require significant improvement to more clearly convey the key messages and strengthen the methodological transparency. I recommend publication after consideration of the comments below.*

Authors: We thank the reviewer for this positive overall assessment of our study.

*Improvement of general paper structure and text.*

*The introduction provides a thorough overview of methane biogeochemistry and modeling approaches, but it currently reads more like a literature review than a focused lead-in to the study. The research gap and specific objectives only become clear after several pages. I suggest condensing the detailed process descriptions (e.g., lines 43–78) and emphasizing earlier why reconciling bottom-up and top-down CH4 estimates is scientifically important and what this study contributes. A shorter, more focused introduction would make the study's novelty clearer to readers.*

*Some specifics:*

*•	There is not enough emphasis on why improving process models in the Arctic specifically is important. This context could appear near the start. Highlighting the magnitude or consequences of Arctic top-down vs. bottom-up discrepancies would strengthen the motivation.*

*•	The discussions of bottom-up and top-down approaches are disjointed, even though their integration is central to the paper.*

*•	You could note that process models are often evaluated against flux or site-level data, but less commonly constrained by atmospheric observations. That contrast nicely sets up the study's novelty.*

*•	The research question appears midway through, before introducing top-down methods. The introduction should build logically from the knowledge gap to the research question, with the final paragraph outlining how the study addresses it.*

Authors: We thank the reviewer for this suggestion. We will revise the introduction to improve its focus and clarity. The first two paragraphs will introduce the Arctic-specific motivation and the importance of reconciling bottom-up and top-down methane estimates. We have made significant cuts to the text to allow focusing on the core issues addressed by the presented study. Most importantly, we have reduced the thorough overview on biogeochemical processes related to the methane cycle to the $CO_2/CH_4$ ratio and the $Q_{10}$-dependence of $CH_4$ production, which are the key parameters investigated in our numerical experiments.

*Similar to the introduction, Section 2.2 is longer than needed and includes detail that is not directly relevant to the study's research question or required to understand the methodology. Much of this description could be replaced with references to existing JSBACH publications or model documentation. For example, lines 146–158 provide an in-depth explanation of inundation and hydrology processes, which are not examined in the subsequent analysis. Parts of the following paragraph (lines 159–178), could be summarized briefly, cited, or moved to a supplement or appendix. As a reader with an inversion rather than process modelling background, I found this paragraph (lines 159-178), difficult to follow. It aims to introduce the relationship between fCH4 and the CH4:CO2. I would suggest revising this section and perhaps including a mathematical relationship between fCH4 and CH4:CO2.*

Authors: We agree that some parts of Section 2.2 could be shortened and referenced more clearly to existing JSBACH documentation. We will revise the text accordingly and make more clear the relationship between fCH4 and CH4:CO2 ratio.

*Discussion and conclusion:*

*As you state, this work provides a valuable initial approach but not a long-term solution -future models will need to allow dynamic changes in $Q_{10}$ to reflect varying environmental conditions. I think this study is timely and also naturally leads into a broader discussion of methane data assimilation techniques, which aim to optimise process-model parameters using observational constraints. For example, Montiel et al. (2025) conduct inversions with a process model (LPJ-GUESS) that has been optimised using eddy covariance fluxes and compare these with simulations using unoptimised parameters, drawing conclusions about the potential of data assimilation. Bernard et al. (2025) also discusses this direction. However, these studies do not yet address regional or seasonal optimization, which your results highlight as particularly important. I think a discussion on this would support the position of your work.*

Authors: We thank the reviewer for highlighting recent advances in methane data assimilation. Following this suggestion, we will add the discussion to place our work in the context of these approaches, including both of the recent studies suggested, Monteil et al. (2025) and Bernard et al. (2025). We will explicitly discuss how these approaches demonstrate the potential of observationally constrained process-model optimization, while noting that regional and seasonal parameter optimization remains largely unexplored.

*In the conclusions section, almost the entire second paragraph focuses on the general limitations of using top-down inversions. This material would fit better in the discussion, where reflection on the limitations of your own inversion setup should be included. Shifting it there would help keep the conclusions concise and focused on the study's main findings.*

Authors: thanks for the suggestion. We will add a new section (Section 3.5 Limitations of Top-Down $CH_4$ estimates) to discuss the limitations of the inverse modeling and make the conclusion shorter and clear.

**Data and code availability**

I notice there is no data or code availability section. Please could this be included for transparency? Could you please properly acknowledge the data providers?

Authors: The datasets assimilated in the inversion framework were already cited in the manuscript, and the corresponding data providers were acknowledged in the Acknowledgements section. To further improve transparency, we will add a dedicated Data and Code Availability section, which clarifies that the prior and posterior fluxes used in this study will be made publicly available upon acceptance of the manuscript. In addition, we will enhance the visibility of the description of the data used in the study.

**Specific comments:**
*Line 36: bottom-up methods also include inventories*
Authors: We will include inventories to this sentence.

*Line 98: "used a prior" or "used as priors"*
Authors: We will edit this.

*Line 102: Do you mean the combined wetlands and inland freshwater sources?*
Authors: Yes, both combined. We will clarify it in the text.

*Line 114: Can you make it clear here or somewhere before how fch4 relates to CO2:CH4.*
Authors: Yes, we will clarify it in the text.

*Section 2.1 Please could you include a list of sites and site years as a table. Potentially in an Appendix?*
Authors: Thanks for the suggestion. We will add a supplementary table (Supplementary Table 1) listing the stations and the years for which data is available.

*Figure 1. Why is there a gap between Alaska and Eastern Russia regions?*
Authors: Thank you for pointing this out. The TM3 model uses a regular latitude-longitude grid. For this study, simulations were carried out at a horizontal resolution of approximately 3.8° and 5° (latitude and longitude), with the -180° longitude point representing a grid cell center rather than a grid boundary. This results in an apparent gap in the visualization between Alaska and eastern Russia, although the model grid itself is continuous.

*Line 140: What is the resolution of the climate data?*
Authors: The CRUJRA2.3 dataset originally had a resolution of 0.5 deg latitude by 0.5 deg longitude, and it was regridded to the JSBACH T63 resolution.

*Line 155: As I mentioned above, I don't think that this level of description is necessary, however here 'the sensitivity study' is mentioned, but I am not sure what that is? Is that in this paper?*
Authors: We will revise this section and remove any details that are not essential to understanding our methodology. With regard to the sensitivity study mentioned in this sentence, we would like to clarify that it does not refer to a sensitivity analysis performed in this paper, but rather to a sensitivity study on the parameterization of JSBACH that was previously performed.

*Line 184: Please can you justify why you chose these values and their limits?*
Authors: The range of $Q_{10}$ values tested in our sensitivity experiments was based on previous studies and literature review. As summarized by Moser et al. (2026), the majority of models

set the temperature sensitivity of CH$_4$ production to be between 1.5 and 4, typically using a central value of around 2.

*Line 188: Since this is a linear Bayesian system, a short description of the cost function or posterior solution would be useful.*

Authors: Thanks for the suggestion. We will review the methods section describing the inverse modelling and add a brief description of the Bayesian cost function and posterior solution to clarify how the inversion is formulated. This addition will clarify that the inversion yields analytical maximum a posteriori flux estimates and associated posterior uncertainties within the linear Bayesian framework.

*Line 196: Are you optimizing by grid cell? By region? Are you running TM3 daily and optimizing daily?*

Authors: Fluxes were optimized by grid cell and fluxes are resolved on a daily time step. As requested by the reviewers, we will add more details about the model setup in section 2.3.

*Line 196: What is the resolution of the meteorology?*

Authors: The meteorology is based on NCEP1 reanalysis data, which has originally a horizontal resolution of 192x94 grid points (longitude x latitude). This corresponds to approximately 1.875° in longitude and 1.915° in latitude. The data include 28 vertical levels, and it was regridded to the TM3 resolution.

*Line 197: This spatial resolution seems quite coarse. Perhaps this could also be discussed in the limitations.*

Authors: In the revised manuscript, we will explicitly discuss the limitations associated with the coarse spatial resolution of the atmospheric inversion in the new section "Section 3.5. Limitations of Top-Down CH$_4$ estimates".

*Line 199: So, is the model data mismatch the same for every site? Are your covariance matrices diagonal, or do you take into account spatial/temporal correlations?*

Authors: The model data mismatch was not the same for all the sites. Each station is assigned a weekly error value based on how well the atmospheric transport model can capture local atmospheric dynamics. For example, mountain sites and stations near shores samples are assigned a smaller error of 15 ppb, whereas surface sites in regions with complex circulation patterns receive a larger error of 30 ppb. We will revise the text to clarify it. In our model, the covariance matrix is diagonal, reflecting combined measurement error, location-dependent modeling error and a data density weighting as described in CarboScope technical report cited in the manuscript (Rödenbeck, 2005).

*Line 200: Please could you provide a short description of the weighting scheme?*

Authors: Yes, thanks for the suggestion. We will add that to ensure balanced representation across observational sites, particularly between continuous and sparse time series, we applied a data density weighting scheme, assigning equal influence to each weekly period, regardless of data frequency as described in Rödenbeck, 2005. Without this adjustment, sites with high-frequency data would dominate the cost function solely because of the greater number of observations. To avoid this, the uncertainty of each measurement is multiplied by the number of observations per week. This corresponds to the assumption that errors are correlated on weekly timescales, meaning that one week of hourly data provides roughly the same amount of independent information as one weekly flask sample.

*Line 202: I think I understand from later in the paper that the emissions are optimized together and not separately (i.e. one scaling for all sources). Please could you make that clear here? Perhaps this could also be discussed in the limitation section.*

Authors: The flux vector $f$ represents the net flux per grid cell per time step. The Jena CarboScope enables $f$ to be represented as the sum of different flux components, each of which is modeled independently using its own statistical linear flux model. These independent a priori error covariance structures allow deviations from the prior flux estimate to be attributed to specific components during the inversion process. In this study, the a priori shape uncertainty was set to 100% of the prior flux for each flux category, and all categories were optimized. Temporal and spatial fluxes are optimized within a Bayesian inversion framework that minimizes a cost function combining prior and observational constraints. This information will be included in the manuscript together with the description of the cost function.

*Lines 202: Please include information on the fire emissions prior! And also, you don't include an inland water/freshwater prior. Please mention this.*

Authors: The fire emissions used as prior fluxes were obtained from the JSBACH model. They are prescribed as monthly-varying biomass burning emissions, as described in Kleinen et al. (2020). This information will be clarified in the manuscript. Additionally, we explicitly state that inland water (freshwater) methane emissions are not included as a separate prior category and are not optimized in the inversion framework.

*Line 225: Taking into account uncertainties in the model/transport etc.*

Authors: We will add that to this sentence.

*Section 2.4: Could this section be broken into multiple paragraphs for readability? Perhaps one for each top-down result (pan-Arctic, seasonal, regional).*

Authors: As suggested, Section 2.4 will be restructured into multiple paragraphs to enhance readability and clarify pan-Arctic, seasonal and regional analyses.

*Figure 2: Figure 2a is a bit fuzzy. Could this be exported as a vector file? Maybe in a and b you could use colors/patterns that nicely show how the results are grouped more by Q10 than they are by fch4. I think the B labelling is not necessary, you also use this in Table 1, but B1_low etc. is not used in the text anywhere.*

Authors: We will improve the resolution of Figure 2a and remove all B labeling from the plots and table. We chose to use a standardized color scheme for all figures to group results by $fCH_4$ and ensure visual consistency, which facilitates comparison between panels and figures throughout the manuscript. Therefore, we prefer kept the color scheme as originally submitted and standardized in the manuscript.

*Figure 3. In c, it's quite difficult to see Alaska, perhaps you could plot this as percentage difference? There is a small typo in the title "difference"*

Authors: As suggest by the reviewer, we will change panel C to show the relative difference between the posterior and prior ensemble fluxes.

*Figure 4. I think these results are really important! Please could you 'zoom-in' on the Arctic-boreal region? Perhaps you could highlight in the legend the experiment that was the best on a pan-Arctic scale, which is clearly not the best regionally.*

Authors: Thanks for the suggestion, we will "zoom-in" on the Arctic-Boreal region and add in the legend the experiment that was the best on a pan-Arctic scale.

References

Monteil, G., Theanutti Kallingal, J. & Scholze, M. CH4 emissions from Northern Europe wetlands: compared data assimilation approaches. Atmospheric Chemistry and Physics 25, 14251–14277 (2025).

Bernard, J. et al. Satellite-based modeling of wetland methane emissions on a global scale (SatWetCH4 1.0). Geoscientific Model Development 18, 863–883 (2025).

---

## Author Comment (AC2)

**Referee 2**

Thanks for your very helpful comments and suggestions. Please find below our answers for each general and technical comment.

*The authors present a study on the evaluation of nine different scenarios of methane (CH4) wetland emissions in the Arctic, obtained by varying two parameters of the JSBACH land surface model: Q10 and fCH4. These scenarios were evaluated using an inverse modeling approach by analyzing the necessary adjustment of the model between prior and posterior CH4 fluxes.*
*The authors found that a Q10 value of 1.8 generally produced the best prior emission scenario in the pan-Arctic region. However, at a regional scale, the optimal parameter set-ups varied, highlighting the importance of using specific parameters of different regions.*
*In my opinion, this study has been well prepared and carefully thought out, and no major adjustments are required. However, there are three aspects of the description of the study set-up and manuscript structure that could be improved:*
Authors: We thank the reviewer for this positive overall assessment of our study.

*1. I would suggest revising the introduction and condensing the information provided slightly. While it is interesting to read, I think it could be shortened slightly to focus more on the research questions being discussed.*
Authors: We will revise and improve the clarity and focus of the introduction, as suggested by the reviewers. We will also emphasize the research questions addressed in this study.

*2. I would also suggest describing the observation network used more extensively, and properly acknowledging the institutions that provided the observations. In my opinion, the terms currently used in the study, such as "different databases" or "several global and regional networks", are insufficient. Additionally, the limitations of the in situ network, such as the lack of observation sites in Siberia, should be discussed earlier in the manuscript, as these can have a significant impact on posterior CH4 emissions.*
Authors: Thanks for the suggestion. We agree with the reviewer about how important is to properly describe and acknowledge the data providers. We will add a more detailed description of the data used (including a supplementary table with a list of stations used in the inversion) and discuss about the limitations of the in situ network.

*3. Please provide a more thorough description of the inversion set-up in section 2.3, as several aspects have not been sufficiently described so far. For example, how did you define the transport error, and which uncertainties were used for the prior emissions? Did you optimize the total CH4 fluxes, or were the fluxes optimized by source category? This is unclear from the description. How were the initial concentrations defined? You could also potentially include one or two more sentences describing the transport model used.*
Authors: We thank the reviewer for this suggestion, and we agree that providing more information about the inversion setup is important for strengthening the paper. We will review the inversion setup in Section 2.3 to better describe the model setup and address the points made by both reviewers. The Jena CarboScope is a linear Bayesian framework that infers surface-atmosphere CH$_4$ fluxes by combining prior flux estimates with atmospheric CH$_4$ mole fraction measurements and accounting for their respective uncertainties based on observed atmospheric mole fractions. The flux vector $f$ represents the net flux per grid cell per time step. The Jena CarboScope enables $f$ to be represented as the sum of different flux components, each

of which is modelled independently using its own statistical linear flux model. These independent a priori error covariance structures allow deviations from the prior flux estimate to be attributed to specific components during the inversion process. In this study, the a priori shape uncertainty was defined as 100% of the prior flux for each flux category. All flux categories were optimized, assuming spatial correlation lengths of ~500 km. Temporal and spatial fluxes are optimized within a Bayesian inversion framework that minimizes a cost function combining prior and observational constraints. The solution is obtained analytically using the linear Bayesian approach, which yields maximum posterior flux estimates and their associated uncertainties. Details of the cost function formulation and solution method can be found in the CarboScope technical report (Rödenbeck, 2005).

In addition, we will also include that the transport model used in CarboScope is the TM3 global atmospheric tracer model, an Eulerian transport model that solves the continuity equation (and parametrizations of boundary layer and convective mixing) for atmospheric tracers in a three-dimensional grid over the globe (Heimann and Körner, 2003). The model has a spatial resolution of approximately 3.8° latitude by 5° longitude, with 19 vertical layers, and it is driven by meteorological inputs from the NCEP reanalysis dataset (Kalnay et al., 1996). Flux inversions were conducted at the TM3 spatial resolution and a daily temporal resolution. Since the model is initialized with a homogeneous background concentration of the tracer, it is run for at least one year before to the period of interest to avoid any impact resulting from the model spin-up. To account for model-data mismatch, including the representation error of the measurements within the transport model, each station is assigned a weekly error value based on how well the atmospheric transport model can capture local atmospheric dynamics. For example, mountain sites and stations near shores samples are assigned a smaller error of 15 ppb, whereas surface sites in regions with complex circulation patterns receive a larger error of 30 ppb.

**Specific comments**

*P1, L17: Would it be possible to already give a short definition of what the Q10 value indicates in the abstract?*
Authors: Yes, we will add it to the abstract.

*P4, L123 and P5, Fig.1: I would consider renaming the "Europe" region "Europe and Greenland", given that Greenland constitutes a substantial part of this region (even though it belongs to Denmark, it is politically independent and not on the European continent).*
Authors: We thank the authors for this suggestion. We will edit the manuscript renaming the "Europe" region to "Europe including Greenland".

*P4, L127: It would be good to mention here that you are using in situ data for the inversion, since "data coverage" could also include satellite data.*
Authors: We will clarify that.

*P6, L168-169: of your choice? Out of curiosity, is the capped fraction of 0.5 a default of the model or a setting*
Authors: It reflects empirical evidence that methanogenesis typically contributes less than half of total anaerobic carbon mineralization, because more energetically favorable anaerobic pathways (e.g., iron, sulfate, and nitrate reduction) generally dominate carbon flow (Bridgham et al., 2013).

Bridgham, S.D., Cadillo-Quiroz, H., Keller, J.K. and Zhuang, Q. (2013), Methane emissions from wetlands: biogeochemical, microbial, and modeling perspectives from local to global scales. Glob Change Biol, 19: 1325-1346. https://doi.org/10.1111/gcb.12131

*P7, L184: How did you define the ranges of Q10 and fCH4? Are these based on experience and/or other studies?*

Authors: The range values tested in our sensitivity experiments was based on previous studies and literature review. As summarized by Moser et al. (2026), the majority of models set the temperature sensitivity of $CH_4$ production to be between 1.5 and 4, typically using a central value of around 2.

*P7 and P8, Section 2.4: Would it be possible to summarize the described calculations for the evaluation in one or multiple equations?*

Authors: We thank the reviewer for the suggestion. Although it is technically possible to summarize the evaluation using equations, our methodology involves ensembles, spatial averaging, and temporal aggregation. Condensing these steps into a few equations could make the workflow more abstract and difficult for readers to follow. Therefore, we chose to present the calculations descriptively to maintain clarity and accessibility.

*P11, L290-L 196: Did the inversion optimize the total CH4 emission or was each emission category optimized separately? In the first case, how were the wetland emission obtained? (See also general comment 3)*

Authors: We optimized each $CH_4$ emission category separately. We will review the Section 2.3 to include a more detailed description of the inversion methodology and to clarify this.

*P13, Figure 3a: So these are the total CH4 emissions from all sources using mean values of all 9 emission scenarios? "using different values of Q10 parameter and baseline fCh4 fraction" is a bit vague and could indicate, that only specific scenarios were used. Also it could be beneficial to plot a pattern in either the prior or the posterior bars since the color difference not always clear (e.g. https://matplotlib.org/stable/gallery/shapes_and_collections/hatch_style_reference.html )*

Authors: We will revise the figure caption to clarify that the values shown represent only the wetland $CH_4$ emissions from all nine inversion scenarios. Additionally, we will improve the figure by adding a pattern to visually distinguish prior from posterior emissions.

*P14, L353-L354: I think it could be helpful to also provide exemplary maps of the prior fluxes (not just the model adjustment) to better visualize expressions such as "which showed that in high-emission areas, for example the Western Siberian Lowlands..."*

Authors: We will add the prior and posterior flux maps as a supplementary figures.

**Technical corrections**

*P3, L84-L85: Please check grammar, e.g. "One big research question now is how high the Q10 value should be for this temperature dependency of the CH4:CO2 production ratio. In order to answer this question, we employ..."*

*P8, L221-L222: Please check grammar, e.g. "Previous studies have used atmospheric inversion models to evaluate different bottom-up estimates and determine which best reproduces observed atmospheric CH4 data..."*

P8, L239: Better: maps were created

*P12, L325: Please check grammar: "...best agreed well with..."*

Authors: Thanks for pointing this out, we will check the grammar and do the technical corrections.